# Focus on the Scapular Region in the Rehabilitation of Chronic Neck Pain Is Effective in Improving the Symptoms: A Randomized Controlled Trial

**DOI:** 10.3390/jcm10163495

**Published:** 2021-08-08

**Authors:** Norollah Javdaneh, Tadeusz Ambroży, Amir Hossein Barati, Esmaeil Mozafaripour, Łukasz Rydzik

**Affiliations:** 1Department of Biomechanics and Sports Injuries, Kharazmi University, Tehran 14911-15719, Iran; njavdaneh68@gmail.com; 2Institute of Sports Sciences, University of Physical Education, 31-571 Krakow, Poland; 3Department of Health and Exercise Rehabilitation, Shahid Beheshti University of Tehran, Tehran 19839-69411, Iran; ahbarati20@gmail.com; 4Department of Health and Sports Medicine, University of Tehran, Tehran 14179-35840, Iran; e.mozafaripour@yahoo.com

**Keywords:** therapeutic exercises, chronic neck pain, scapular dyskinesia

## Abstract

Chronic neck pain is a common human health problem. Changes in scapular posture and alteration of muscle activation patterns of scapulothoracic muscles are cited as potential risk factors for neck pain. The purpose of this study was to compare the effects of neck exercise training (NET) with and without scapular stabilization training (SST) on pain intensity, the scapula downward rotation index (SDRI), forward head angle (FHA) and neck range of motion (ROM) in patients with chronic neck pain and scapular dyskinesia. A total of sixty-six subjects with chronic neck pain and scapular dyskinesia were randomly divided into three groups: neck exercise training, n = 24, combined training (NET + SST), n = 24 and a control group, n = 24. Pain intensity, SDRI, FHA and ROM were measured by the numerical rating scale, caliper, photogrammetry and IMU sensor, respectively. When the combined intervention group consisting of NET and SST was compared with NET alone at six weeks, there was a statistically significant difference in pain intensity, SDRI, FHA and cervical ROM for flexion and extension (*p* ≤ 0.05). Adding scapular exercises to neck exercises had a more significant effect in decreasing pain intensity, SDRI, FHA and increased cervical ROM than neck exercises alone in patients with chronic neck pain. These findings indicate that focus on the scapular posture in the rehabilitation of chronic neck pain effectively improves the symptoms.

## 1. Introduction

Neck pain that lasts for three months or more is determined as chronic neck pain. The mechanism of nonspecific neck pain is still not clearly understood. While neck pain as etiology is multifactorial and includes working conditions, sedentary lifestyle, postural abnormalities, previous trauma to the neck region and altered neuromuscular control of cervical muscles are the main risk factors for nonspecific neck pain stated in the literature [1]. Changes in scapular posture and muscle activation patterns are cited as potential risk factors for chronic neck pain (CNP) [1]. Subjects with chronic neck pain tend to have more protracted shoulders compared with asymptomatic issues [2]. An altered kinematic of the scapula may be present in subjects with chronic neck pain, which can play a substantial role in the maintaining or intensifying of symptoms in these patients [3,4]. The underlying mechanisms in the relationship between altered scapular kinematics and CNP may be due to changes in the length–tension relationships of muscles that connect the scapula, head, cervical spine and chest. Altered behaviors of muscles, such as the trapeziuses, levator scapula and rhomboid minor, which are directly connected to the cervical spine, may cause compression and shear forces on the neck area and cause pain in this region [4].

Scapular downward rotation syndrome (SDRS) is a common scapular alignment impairment. It is reported that, most often, SDRS typically leads to shortened levator scapula (LS), and lengthened upper trapezius (UT) and serratus anterior (SA) muscles [5]. Furthermore, the lower trapezius (LT) weakness can play a substantial role in insufficient scapular upward rotation [5]. The scapulothoracic muscles play an essential role in transferring the load between the upper limb and the spine [6]. Studies have shown a change in the recruitment of muscle patterns in subjects with neck pain compared to healthy subjects. Individuals with chronic neck pain have different muscle activation patterns and kinematics than individuals without a disorder [1,7]. Subjects with neck pain have less muscle strength and activity than healthy people [8,9]. Pain sensitivity and axioscapular muscle activity have altered in neck pain patients compared with healthy controls [10]. Researchers have shown a relationship between decreased muscle strength and endurance with chronic nonspecific neck pain [11]. Neck stabilization exercises have been popular for managing and preventing spinal dysfunction by recruiting local muscles and regulating the over-activity of surface muscles [12,13]. A systematic literature review on the influence of exercise intervention for chronic neck disorders showed that exercise training has an essential role in the cure of neck pain but stated that more studies are needed to examine the effect of each type of exercise [14].

Even though many studies have been done on chronic neck pain, little is known about the potential benefits of scapular exercise on chronic neck pain [6,15]. According to a systematic review, scapular exercise may improve symptoms of neck pain, but the effects of scapular exercise on pain and dysfunction in the neck region remain unclear because the number of studies was small and recommended that further high quality research is needed [16].

Some studies have used these exercises in combination to treat people with chronic neck pain [15,17,18,19]. Most of the above assignments have not considered the role of scapula condition in treating neck pain, and postural variables have been less studied. Furthermore, due to the lack of comparison between these training interventions, the effect size of this exercise in each region is not known. Scapular dyskinesia also needs to be considered during the management of chronic neck pain. Rehabilitation exercises that aim to return functionality of the scapular muscles are deemed necessary to render a successful result. Therefore, the purpose of this study is to evaluate the effectiveness of adding the scapular stabilization training to the neck exercise training on pain, the scapula downward rotation index, forward head angle and neck ROM in the patient’s chronic neck pain with scapular dyskinesia. We hypothesized that adding scapular stabilization training to neck exercise training will increase treatment efficacy on these variables. 

## 2. Materials and Methods

### 2.1. Study Design

This study was a three-arm randomized control trial, with two intervention groups and a control group, and was conducted according to the Consolidated Standards of Reporting Trials (CONSORT) guidelines [15] and registered at UMIN-CTR Clinical Trial (ID: UMIN000043938). A total of 72 patients were recruited from two rehabilitation and physiotherapy centers between May 2020 and October 2020 in Tehran city. All expected outcomes were collected at the Noor Health Center. The study was conducted following the Helsinki Convention and approved by the Ethics Committee of the Sports Science Research Center (ID: IR.KHU.REC.1398.011). Written informed consent was obtained from each patient to be included in this study. Participants were assessed before the study and after 6 weeks of intervention (end of the exercise intervention) by a physiotherapist blinded to the participants’ groups. An external assistant physiotherapist, blinded to the participants’ allocation groups, was responsible for collecting patient data. The independent variables were neck exercise, combined exercise (neck exercise + scapular stabilization exercise), control group and time (pre-intervention, post-intervention). The dependent variables were pain intensity, scapula downward rotation index and forward head angle.

### 2.2. Participants

Patients with chronic neck pain were recruited through a text message on social networks and via flyers displayed at the hospitals. They were selected based on the eligibility criteria listed below. A total of 72 patients with ongoing chronic neck pain volunteered for this study. In this study, chronic neck pain was identified as neck pain with no specific cause, such as inflammation, disease and infection, but was stimulated by palpation, and neck movement [20].

Inclusion criteria involved people between 20 and 50 years of age who had a history of ongoing bilateral neck pain for three months or more. Furthermore, moderate pain intensity (between 3 and 7 based on VAS) and having bilateral scapula downward rotation (participants had to score 5 mm or above on the base scapula downward rotation index) were among the inclusion criteria. Exclusion criteria were any previous shoulder or neck surgery, fibromyalgia and pathology and having a poor general health status that would interfere with the exercises during the study. The inclusion/exclusion criteria were confirmed by a physician and three physiotherapists, by history and physical examination.

Sample size calculations using G*Power software (v3.1.9.2, Heinrich-Heine-University, Dusseldorf, Germany) as in the previous studies [21,22] resulted in 66 patients (22 patients per group). Considering an effect size of 0.23, a statistical power of 0.8%, and an alpha of 0.05 (two-tailed test), a total sample size of 66 was required (22 patients per group). An allowance was made for a 10% dropout rate, increasing the sample size to 72 patients (24 per group). Patients were randomized by the slot-drawing method into one of three groups: group (1) neck exercise training; group (2) combined (neck exercise training plus scapular stabilization training); group (3) control group (Figure 1). The allocation was by sealed opaque envelopes, and patients were assigned to each group by a sealed envelope containing one of the three groups. For the allocation of participants, computerized random numbers were used. In this study, the assessor was blinded; however, patients were aware of what treatment they were participating in. The inclusion/exclusion criteria and outcome measurements were assessed by an orthopedic surgeon and physiotherapist, blinded to the study’s procedures. The outcome assessors and data analysts were kept blinded to the group allocation to intervention or control group.

### 2.3. Outcome Measure(s)

Pain intensity was measured with the Visual Analogue Scale (VAS). Patients were instructed to assess the severity of neck pain experienced last week on a 0–10 cm horizontal line (0 = painless and 10 = worst pain imaginable) [23]. The VAS has been shown to have excellent test–retest reliability (ICC = 0.97) and high validity (r with a 5-point verbal descriptive scale = 0.71–0.78) to evaluate pain perception. An alteration of two points or more was identified as the minimal clinically important difference in patients with chronic neck pain [23].

The scapula downward rotation index (SDRI) and forward head angle were measured by caliper and photogrammetry, respectively. The modified Kibbler method was used to measure the SDRI and was conducted on the dominant hand side using a caliper. The SDRI was calculated using the following equation: the distance between the second thoracic vertebra and spine of the scapula minus the distance between the seventh thoracic vertebra and the inferior angle of the scapula. Positive values demonstrated downward rotation scapula [21,24]. The interclass correlation coefficient (ICC) of the inter-rater reliability was 0.85 and the ICCs of the intra-rater reliabilities were 0.88–0.96 [21,24].

Forward head posture was measured using photogrammetry (Canon PowerShot, SX130IS). This is the angle composed at the intersection of a horizontal line through the seventh cervical vertebra and a line to the tragus of the ear. The craniovertebral angle (CVA) was analyzed using postural assessment software (Sony Cyber-shot DSC-P93, Sony, San Jose, CA, USA). A craniovertebral angle less than 48°–50° is defined as forward head posture [25]. Craniovertebral angles have been proven to be valid measures of posture when compared with similar angles measured on radiographs [25].

The MyoMotion (Noraxon Inc., Scottsdale, AZ, USA) 3D motion analysis system was used to investigate the cervical ROM. A small inertial measurement unit (IMU) sensor placed on a body segment tracked 3D angular orientation. The IMU 3D motion analysis is acceptable in its validity and reliability for the cervical ROM [26]. The MyoMotion IMU sensors include a 3D accelerometer, gyroscope and magnetometer, and when placed on a body segment, can be used to determine the segment’s three-dimensional orientation. The IMU 3D motion analysis is completely wireless and does not require calibration. For the cervical ROM assessment, an IMU sensor was pasted to the head (middle of the front of the head) using a flexible and adjustable strap. The cervical ROM changes were recorded with the sampling frequency at 200 Hz [26]. The calibration posture was sitting straight with neutral head positioning and the arms next to the body with the elbows bent at 90° to determine the value of the 0° angle in the cervical joint [26]. Data were analyzed using the Noraxon MyoResearch 3.14.32 Windows software (Noraxon Inc., Scottsdale, AZ, USA). The patient performed flexion and extension movements in the sagittal plane. The ICC of each movement was over 0.8.

### 2.4. Rehabilitation Interventions

Neck exercise training (NET): The composition and progress of the exercises were designed according to the exercises presented in previous studies [22,27,28,29]. The exercises included craniocervical flexion using the stabilizer pressure biofeedback unit, neck isometric exercises using Thera-band and neck stability exercises in supine, prone, quadrupedal and bipedal positions. Further details on the exercise protocol are reported in the study by Javdaneh et al. [22]. NET was implemented once a day (three days per week, for six weeks). More details on the interventions and exercise are reported in Table A1.

Scapular stabilization training (SST): Scapular stabilization exercise training covered exercises for the muscles influencing scapular alignment related to chronic neck pain [21]. The composition of the SST was planned based on prior research [21]. The progressive SST training was designed based on sports medicine principles [30]. Exercises included: non- resistive SUR exercise [31], wall facing arm lift, prone arm lift, backward rocking arm lift [32], elevation of the arm in line with the lower trapezius muscle fibers, elevation of the arm in the plane of the scapula [33], shoulder shrug, [34] and stretching of levator scapular and the pectoralis minor muscle [35]. The exercises were performed three days per week, for six weeks (18 sessions in total). More details on the interventions and exercise are reported in Table A2.

The exercise training was implemented under the surveillance of three physiotherapists and an athletic trainer, who had more than five years of experience in the treatment of the musculoskeletal system. Each training session took 40–60 min. It was composed of 10 min of warm-up exercises, 30 min of scapular exercises and a 5 min cool-down.

The control group participated in a session in which they were taught a home exercise program that mainly focused on the posture of the body during daily work and demonstrations of lifting, pressing, pulling tasks and office ergonomics. After the end of the interventions, the control group received a comprehensive rehabilitation program.

### 2.5. Statistical Analysis

The Statistical Package for the Social Sciences (SPSS, IBM Corporation, Armonk, NY, USA) version 19.0 was used for statistical analysis. A Shapiro–Wilk test was performed to test the normality of the data. The variance of repeated measures (RM-ANOVA) was used to examine the differences between groups. Effect sizes were calculated and were interpreted according to Cohen d (trivial < 0.2, small = 0.2–0.5, medium = 0.5–0.8 or large > 0.8). Mathematically, Cohen’s effect size is denoted by; d = M_1_ − M_2_/S. A level of 0.05 was identified as being of statistical significance.

## 3. Results

One hundred fifteen subjects were screened and 72 were selected and randomized after consideration of the inclusion and exclusion criteria. Five patients withdrew from the study due to personal reasons before completing the interventions (two from the neck exercise group, two from the combined group and one person from the control group). There was a high degree of adherence to the two interventions (18 sessions for both intervention groups). No adverse events were reported. The baseline and demographic characteristics of the 72 patients included in the study are reported in Table 1. At the baseline, there were no significant differences between groups in any of the demographic characteristics and clinical variables. Finally, the data of 67 subjects were analyzed after the intervention.

The results showed a significant effect of time (*p* < 0.001), group (*p* < 0.001) and time by group interaction (*p* < 0.001) for VAS, ROM, SDRI and FHP. Significant differences between groups were found for VAS after the intervention. Reduction in the severity of pain (VAS) was significantly higher in the combined group (neck exercise and scapular stabilization exercise) than the neck exercise alone and control groups. For VAS, between neck exercise intervention vs. combined intervention (effect size (ES) = −2.71, *p* = 0.001), neck exercise vs. control (ES = 4.16; *p* = 0.001) and combined group vs. control (ES = 5.45; *p* = 0.001) significant differences were observed (Table 2).

Significant differences between groups were found for the scapular downward rotation index and forward head angle after the interventions. For the scapular downward rotation index, differences between the neck exercise training group vs. combined group (ES = −3.56, *p* = 0.001), and combined intervention vs. control (ES = −4.55; *p* = 0.001) were observed as significant, and in neck exercise vs. control (ES = 0.74; *p* = 0.43) no significant difference was observed. For forward head angle, differences between the neck exercise training group vs. combined group (ES = −1.86, *p* = 0.001), neck exercise vs. control (ES = −1.23; *p* = 0.001) and combined intervention vs. control (ES = −3.09; *p* = 0.001) were observed to be significant (Table 2).

Significant differences between groups were found for neck flexion and extension ROM at six weeks. For neck flexion ROM, differences between the neck exercise training group vs. combined training (ES = 1.54, *p* = 0.024), combined training vs. control (ES = 3.47; *p* = 0.001) and also combined training vs. control (ES = 1.90.; *p* = 0.01) were observed to be significant. For neck extension ROM, differences between neck exercise training vs. combined training (ES = 1.57, *p* = 0.025), neck exercise training vs. control (ES = 3.07; *p* = 0.001) and also combined training vs. control (ES = 4.06; *p* = 0.001) were observed as significant (Table 2).

## 4. Discussion

The results of this study indicated that VAS, SDRI and FHA decreased in neck exercises training alone (except SDRI) and in combined intervention in subjects with CNP and scapular dyskinesia. The results also showed a significant increase in the cervical ROM after the intervention. This study revealed that a combined intervention group including neck exercise training and scapular stabilization exercise training was superior to neck exercise training alone in improving the variables of subjects with CNP.

The results of this study are relative to improvements in VAS compliance with prior studies investigating the effects of neck exercise training [22,28]. The mechanism through which neck exercise training reduces chronic neck pain may be based on the notion that exercise training increments activity in the motor pathways, thereby exerting an inhibitory effect on pain receptors in the central nervous system [36]. Furthermore, it may be that the development in neuromuscular control from neck exercise training decreases the compression placed on the joints. The afferent input induced by exercise applications may stimulate neural inhibitory systems at various levels in the spinal cord and activate descending inhibitory pathways from the midbrain and decrease pain [37].

The results of this study are consistent with the results of previous studies. Thompson et al. showed a positive impact of a progressive neck exercise program, including isometric strengthening exercises of the cervical flexors, extensors and side flexors, upper limb strengthening exercises and cervical stretching exercises on people with chronic neck pain [18]. Ylinen et al. showed that both strength and endurance training, including dynamic neck exercises of the neck, shoulders and upper extremities, were effective methods for decreasing pain and disability in women with chronic, nonspecific neck pain [19].

Correcting the scapula position passively or actively has been shown to decrease chronic neck pain [38,39]. Combination therapies are recommended due to better pain intensity than manual therapy or exercise alone [40]. It was demonstrated that the correction of the scapular position may decrease the tension in the axioscapular muscles and abnormal cervical loads, and therefore the pain decreases [6]. Another reason for reduced pain is the reduction of downward pulling tension due to special exercises for upward rotation muscles of the scapula. The downward pulling tension due to sustained scapular downward rotation defects arose in participants. The sustained compressive force of posterior neck spine can induce pain by a downward pull on the neck spine or facets via the levator scapula and upper trapezius muscles. The exercise training may have altered the tension–length relationship of the tightened scapular downward rotator muscles, causing the reduction in downward pulling tension, and consequently decreasing the force on the neck facet joints [41]. The main action of the levator scapulae muscle is to elevate and downwardly rotate the scapulae. With its connection to the upper four neck vertebrae and vertically oriented muscle fibers position, this condition puts undesirable pressure on the upper part and a potential compressive force on the lower part of the cervical vertebrae. Therefore, decreased levator scapulae muscle activity may help to reduce shear force and compressive load on the neck region during active neck movements [2] and thus reduce pain intensity.

The scapula exercise in the current study focused on increasing activation of the upward rotation muscles of the scapula. The serratus anterior, upper trapezius and lower trapezius muscles are considered as scapular upward rotator muscles. These exercises may have caused a decrease in the imbalance between these muscles. Improving the function and strength of the weak muscles and the balance between the scapular upward and downward rotator muscles can improve scapular alignment. By increasing hyperextension and cervical lordosis, the upper trapezius could generate forward head posture. The intervention should manage the extreme activity of the upper trapezius muscle to correct forward head posture and decrease neck pain [42]. Therefore, scapula exercises combined with other interventions may reduce the activity of the upper trapezius, improving scapular alignment and proper scapular rhythm, and ultimately affecting the head-forward angle.

In both programs of the present study, training also significantly increased the neck flexion and extension ROM; however, this increase was substantially higher in the combined group. Altered scapular posture has been detected in subjects with NP, which may be an effective mechanism for intensification of symptoms in these patients [1,43]. Scapular downward rotation syndrome can participate in long comparative loading of the neck region via the transition of the weight of the upper extremities to the cervical spine [3,4]. As observed, exercise therapy reduced pain and it is likely that this reduction in pain will release the muscles from tension and allow the joint to move more.

This study also has several limitations. First, we did not evaluate the long-term effect of the six-week neck exercise with scapular exercise intervention. Even though the six-week neck exercise with scapular exercise intervention is effective on symptoms in the patients with neck pain and scapular dyskinesia, the results may not be generalized for the longer term. Second, in this study, we used patients with scapular downward rotation defects, so other forms of scapular dyskinesia or healthy subjects may have different results. Therefore, this limitation makes it difficult to generalize the results. Finally, we recruited the patients for six weeks; therefore, a longitudinal study of the long-term effects with follow-up is necessary.

## 5. Conclusions

The results suggest that neck exercise training combined with the scapular stabilization exercise was more effective for reducing pain, forward head angle, scapula upward rotation and increasing cervical ROM in patients’ chronic neck pain with scapular downward rotation defects. These findings indicate that focus on the scapular posture in the rehabilitation of chronic neck pain effectively improves the symptoms.

## Figures and Tables

**Figure 1 jcm-10-03495-f001:**
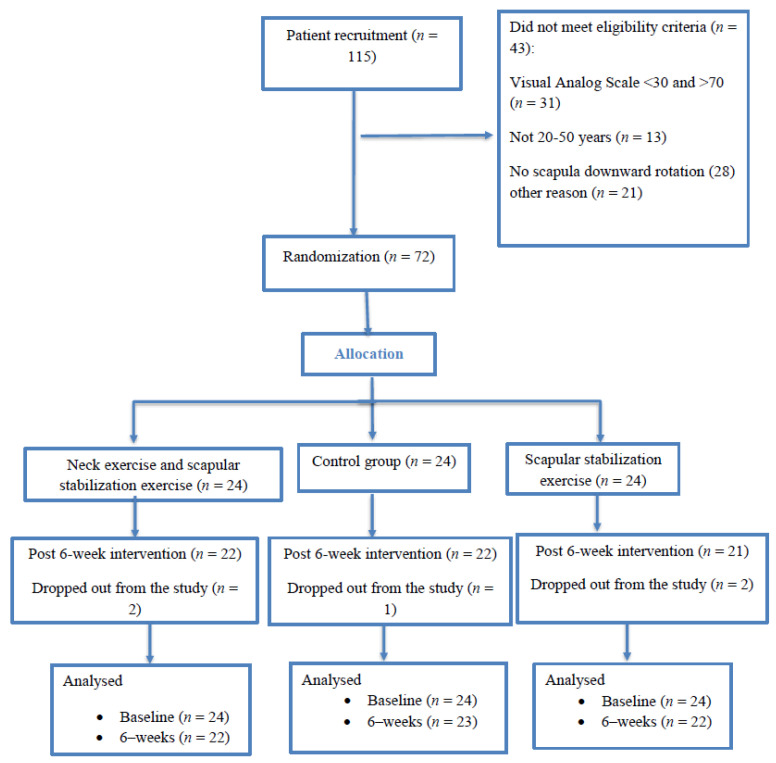
CONSORT flow diagram of the study.

**Table 1 jcm-10-03495-t001:** Demographic characteristics of the participants for all groups at baseline. All values are expressed as mean ± SD.

Variables		Groups (No.)			*p*	Gender Differences
		NET (*n* = 24)	Combined (*n* = 24)	Control (*n* = 24)		*p*
Age (year)		32.58 ± 6.37	34.25 ± 8.01	35.41 ± 7.77	0.77	0.84
Weight (kg)		75.65 ± 4.10	78.5 ± 5.00	76.23 ± 4.05	0.73	0.078
Height (cm)		178 ± 5.20	179 ± 5.28	177 ± 5.68	0.74	0.001
BMI (kg/m^2^)		24.20 ± 2.17	24.33 ± 2.07	24.08 ± 2.05	0.83	0.017
Duration of symptoms (year)		3.18 ± 1.54	4.25 ± 1.85	3.40 ± 1.104	0.57	0.69
VAS at baseline (0–100)		57.15 ± 6.33	59.70 ± 6.15	58.65 ± 6.44	0.83	0.91
SDRI at baseline		1.5 ± 0.13	1.59 ± 0.17	1.59 ± 0.14	0.91	0.80
FHA at baseline		40.50 ± 2.88	39.87 ± 2.90	39.79 ± 2.85	0.87	0.76
Neck flexion ROM at baseline		46.37 ± 4.76	46.33 ± 4.97	45.62 ± 5.40	0.80	0.71
Neck extension ROM at baseline		37.83 ± 4.76	37.58 ± 5.35	37.58 ± 5.35	0.85	0.68
Gender (*n*)	Female	13	14	14	0.67	
	Male	11	10	10		

NET: neck exercises training; Combined: neck exercises + scapular stabilization training; BMI: body mass index; VAS: visual analogous scale; SDRI: scapula downward rotation index; FHA: forward head angle; ROM: range of motion.

**Table 2 jcm-10-03495-t002:** VAS, SDRI, FHA and ROM scores differences between groups.

Variables	Group	Pre-Training ^a^	Post-Training ^a^	Between- Groups Difference (Bonferroni Post-Hoc Test)					
				NET vs. Combined		NET vs. Control		Combined vs. Control	
				Mean Difference (%95 CI)	ES (*p*-Value)	Mean Difference (%95 CI)	ES (*p*-Value)	Mean Difference (%95 CI)	ES (*p*-Value)
Pain, 0–100 (mL)	NET	57.15 ± 6.33	31.55 ± 5.15	8.32 (5.32, 12.45)	2.71 (0.001) *	−13.77 (−17.25, −9.25)	4.16 (0.001) *	−22.10 (−27.1,−17.2)	6.66 (0.001) *
	Combined	59.70 ± 6.15	12.35 ± 4.74						
	Control	58.65 ± 6.44	57.60 ± 6.54						
SDRI	NET	1.5 ± 0.13	1.40 ± 0.14	0.257 (0.12, 0.39)	3.56 (0.001) *	−0.084 (−0.22, 0.05)	0.74 (0.430)	−0.342 (−0.47, −0.20)	−4.55 (0.001) *
	Combined	1.59 ± 0.17	0.87 ± 0.21						
	Control	1.59 ± 0.14	1.56 ± 0.13						
FHA	NET	40.50 ± 2.88	44.54 ± 3.12	−2.12 (−3.93, −0.31)	1.86 (0.001) *	2.85 (1.04, 4.66)	−1.23 (0.001) *	4.97 (3.17, 6.67)	−3.09 (0.001) *
	Combined	39.87 ± 2.90	49.41 ± 3.06						
	Control	39.79 ± 2.85	39.54 ± 3.24						
Neck flexion ROM	NET	46.37 ± 4.76	59.16 ± 5.32	−3.83 (−7.25, −0.40)	1.54 (0.024) *	5.66 (2.24, 9.09)	1.90 (0.001) *	9.50 (6.07, 12.9)	3.47 (0.001) *
	Combined	46.33 ± 4.97	66.87 ± 4.27						
	Control	45.62 ± 5.40	48.58 ± 6.04						
Neck extension ROM	NET	37.83 ± 4.76	52.83 ± 6.10	−3.92 (−7.53, −0.43)	1.57 (0.025) *	7.64 (4.08, 11.20)	3.07 (0.001) *	11.62 (8.06, 15.18)	4.06 (0.001) *
	Combined	37.58 ± 5.35	61.04. ± 7.18						
	Control	37.58 ± 5.35	37.45 ± 5.15						

NET: neck exercises training; Combined: neck exercises + scapular stabilization training; ES: effect size; SDRI: scapula downward rotation index; FHA: forward head angle; ROM: range of motion; ^a^ mean ± standard deviation, * statistically significant difference (*p* < 0.05).

## Data Availability

The datasets analyzed during the current study are available from the corresponding author on reasonable request.

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
