# Peer review of "Focus on the Scapular Region in the Rehabilitation of Chronic Neck Pain Is Effective in Improving the Symptoms: A Randomized Controlled Trial"

_jcm, 2021, doi:10.3390/jcm10163495_

Round 1

Reviewer 1 Report

jcm-1285571. Focus on the Scapular Region in the Rehabilitation of Chronic neck Pain is Effective in Improving the Symptoms.

Dear editors,

I appreciate the opportunity to review this manuscript. It is a basic topic for physiotherapists; therefore the importance of this review. My comments are added below.

Throughout the text, review why there are extra spaces. The words on the boards are cut off. Please, review this

The most important aspect to improve is the design of the methodology.

In the design of the ECA, do you take into account the CONSORT standards? If so, add the CONSORT checklist.

Inclusion criteria are unclear. How was having CNP and bilateral downward rotation of the scapula measured? Please add more detail.

No flow chart diagram.

The authors said “. At the baseline, there were no significant differences between groups in any of the demographic characteristics and clinical variables. The only variable that Table 1 contributes is pain intensity. What about the other clinical variables?

Once the design aspects have been solved, the manuscript can be considered for publication.

Author Response

Dear reviewer

We very much appreciated your encouraging and insightful comments. We have endeavored to respond to all suggestions and comments, which further improved the understanding and potential impact of our manuscript. Detailed responses are given below. In case of further queries, we are happy to clarify any further details and look forward to your reply.

  • Throughout the text, review why there are extra spaces. The words on the boards are cut off. Please, review this
  • Response: revision done.
  • The most important aspect to improve is the design of the methodology.
  • Response: revision done.
  • In the design of the ECA, do you take into account the CONSORT standards? If so, add the CONSORT checklist.
  • Response: This study was a three-arm randomized control trial, with two intervention groups and a control group, and was conducted according to the Consolidated Standards of Reporting Trials (CONSORT) Statement, and registered at UMIN-CTR Clinical Trial (ID: UMIN000043938).Checklist added.
  • Inclusion criteria are unclear. How was having CNP and bilateral downward rotation of the scapula measured? Please add more detail.
  • Response: bilateral scapula downward rotation and bilateral neck pain were the inclusion criteria. Participants had to have bilateral scapula downward rotation and had to score 5mm or above on the based scapula downward rotation index.
  • No flow chart diagram.
  • Response: Flowchart added.
  • The authors said “. At the baseline, there were no significant differences between groups in any of the demographic characteristics and clinical variables. The only variable that Table 1 contributes is pain intensity. What about the other clinical variables?
  • Response: Other variables were added to the table1.

Reviewer 2 Report

I have added all the comments in the text. It is an interesting and pleasant text to read. I think they must modify some things.

Author Response

Dear reviewer

We very much appreciated your encouraging and insightful comments. We have endeavored to respond to all suggestions and comments, which further improved the understanding and potential impact of our manuscript. Detailed responses are given below. In case of further queries, we are happy to clarify any further details and look forward to your reply.

  • Response: All suggestions about the English language and style were applied in the text.
  • Q: I think there is a lack of information on when the recruitment began and when it ended
  • Response: patients were recruited from two rehabilitation and physiotherapy center between May 2020 and October 2020 in Tehran city.
  • Q: I would add information on when the assessment and the second assessment are performed. I would add information on who makes the assessment. Add information on where the assessments are made.
  • Response: All expected outcomes were collected at the Noor Health Center. Participants were assessed at baseline and six weeks (end of treatment) by a physiotherapist blinded to the participants’ groups. An external assistant physiotherapist, blinded to the participants’ allocation groups, was responsible for collecting patient data.
  • Q: I think there is a lack of information about who directs the exercises, time spent with the subject to teach them and how much experience they have in this field
  • Response: The exercise training(Both intervention groups) was implemented under the surveillance of three physiotherapists and an athletic trainer‎, who they had more than five years of experience in the treatment of musculoskeletal system. All patients received individual supervision by a physiotherapist during all sessions. Each exercise session took 40-60 min. It was composed of 10 min warm-up exercises, 30 min exercises of scapular, 5 min cool-down.
  • Q: Add some information about the reason why there was no type of change in the subjects of the control group
  • Response: failure to change the variables in the control group is probably due to insufficient training time and also disregard for instructions during activities.

Reviewer 3 Report

Although I read the manuscript with great enthusiasm, there are some major and minor concerns with the work that I feel should considered. I refer the authors to specific comments below.

Minor points

The title should reflect the study design.

According with the guidelines of the journal, abstracts should be written in a single paragraph without subheadings. Furthermore, p values should be reported in results.

Line 33: Non-specific neck pain is a classification of neck pain regarding the etiology. It should be clarified

Major points

2.1 Study Design: Even if this study followed the Helsinki considerations, an ethical committee approval is mandatory.

RCT are recommended to follow the CONSORT guidelines.

2.2 Participants

Could you state the recruitment time frame?

I do not understand clearly why participants are considered blinded

2.3 Outcomes measures

I recommend the authors to provide utility estimates for all the measurements tools used (validity, reliability, specificity and sensitivity).

Including figure of the MyoMotion tool would be interesting

2.4 Statistical analysis

How was the effect size calculated?

Results:

Why the authors selected 66 participants if the available sample was 115? If the reason for not including those 49 subjects was that they did not meet the eligibility criteria, the reasons should be reported.  If they were excluded to adjust the study to the 66 subjects needed according with the sample size calculation, this is an important limitation of the study since follow-up losses were not considered and finally the minimum sample size was not reached (it is common to increase a 10-15% the minimum sample size required to avoid this problem). A flowchart would be useful.

In table 1, it would be interesting if the demographic characteristics were reported as total sample, by group (already done) and by gender (if there are gender differences, it should be considered). Analysis of Covariance (ANCOVA) using baseline values as covariates has been shown to be more powerful than repeated measures analysis of variance (ANOVA) when random group assignment is used.

Author Response

Dear reviewer

We very much appreciated your encouraging and insightful comments. We have endeavored to respond to all suggestions and comments, which further improved the understanding and potential impact of our manuscript. Detailed responses are given below. In case of further queries, we are happy to clarify any further details and look forward to your reply.

  • The title should reflect the study design.
  • Response: Focus on the Scapular Region in the Rehabilitation of Chronic neck Pain is Effective in Improving the Symptoms: A randomized controlled trial
  • According with the guidelines of the journal, abstracts should be written in a single paragraph without subheadings. Furthermore, p values should be reported in results.
  • Response:
  • Line 33: Non-specific neck pain is a classification of neck pain regarding the etiology. It should be clarified
  • Response: corrected. The mechanism of Nonspecific neck pain is still not clearly understood. While Neck pain a etiology is multifactorial and includes working conditions, Sedentary lifestyle, postural abnormalities, previous trauma to the neck region, and altered neuromuscular control of cervical muscles are the main risk factors for Nonspecific neck pain stated in the literature
  • 1 Study Design: Even if this study followed the Helsinki considerations, an ethical committee approval is mandatory.
  • Response: Ethics committee added.
  • RCT are recommended to follow the CONSORT guidelines.
  • Response: This study was a three-arm randomized control trial, was conducted according to the Consolidated Standards of Reporting Trials (CONSORT) Statement.
  • Could you state the recruitment time frame?
  • Response: patients were recruited from two rehabilitation and physiotherapy center between May 2020 and October 2020.
  • I do not understand clearly why participants are considered blinded
  • Response: Patients were unaware that there were three types of intervention, but patients were aware of what treatment they were participating in.
  • I recommend the authors to provide utility estimates for all the measurements tools used (validity, reliability, specificity and sensitivity).
  • Response: Some items were added. VAS: The VAS has been shown to have excellent test-retest reliability (ICC = 0.97) and high validity (r with a 5-point verbal descriptive scale = 0.71-0.78) to evaluate pain perception. An alteration of two points or more was identified as the minimal clinically important difference in patients with chronic neck pain. SDRI: The interclass correlation coefficient (ICC) of the inter-rater reliability was .85 and the ICCs of the intra-rater reliabilities were .88-.96. ROM: The IMU 3D motion analysis is acceptable validity and reliability for cervical range of motion. FHP: Craniovertebral angle have been proved to be valid measures of posture when compared with similar angles measured on radiographs.

  • How was the effect size calculated?
  • Response: Effect sizes calculated and were interpreted according to Cohen d (trivial <0.2, small = 0.2–0.5, medium = 0.5–0.8, or large >0.8). Mathematically Cohen’s effect size is denoted by; d=M1-M2/S.
  • Why the authors selected 66 participants if the available sample was 115? If the reason for not including those 49 subjects was that they did not meet the eligibility criteria, the reasons should be reported. If they were excluded to adjust the study to the 66 subjects needed according with the sample size calculation, this is an important limitation of the study since follow-up losses were not considered and finally the minimum sample size was not reached (it is common to increase a 10-15% the minimum sample size required to avoid this problem). A flowchart would be useful.
  • Response: Details were added in Flowchart 1. Sample size calculations using G*Power software resulted in 66 patients (22 patients per group).An allowance was made for a 10% drop-out rate, increasing the sample size to 72 patients (24 per group).
  • In table 1, it would be interesting if the demographic characteristics were reported as total sample, by group (already done) and by gender (if there are gender differences, it should be considered). Analysis of Covariance (ANCOVA) using baseline values as covariates has been shown to be more powerful than repeated measures analysis of variance (ANOVA) when random group assignment is used.
  • Response: By examining the results, no gender differences were observed for the dependent variables

Round 2

Reviewer 2 Report

I note that the appropriate modifications have been made, and the English have been revised. Now, in my opinion, it is suitable to publish

Author Response

Thank you

Reviewer 3 Report

Thank you for your kind reply to the comments given.

Abstract: ≤ should be changed for < (if applicable)

Introduction: Although the authors explained the etiology of neck pain, in the first sentence “nonspecific” should be removed since chronic neck pain refers the duration and nonspecific refers the etiology.

Methods: Even if the Institutional Ethics Committee has been named, is necessary to include the identification number or date of approval. In addition, this should be added in line 342-343

CONSORT is a guideline, not statement. Reference is necessary

“In this study, the assessor and patients were blinded, but patients were aware of what treatment they were participating in”. In my opinion and based on this sentence, this study is not double-blinded since participants were aware of the treatment group.

Results: My request was not just to provide the number/percentage of males and females in each group. A new column with two groups (males and females) should be added reporting sociodemographic and clinical characteristics and analyzing if there are between-genders difference by using a Student-T test if variables are normal distributed or non-parametric test if not.

If differences are found, a Bonferroni post hoc analysis (Group*Gender) is needed.

Author Response

Dear reviewer, we very much appreciated your encouraging and insightful comments.

Abstract: ≤ should be changed for < (if applicable):

The abstract became shorter.

Introduction: Although the authors explained the etiology of neck pain, in the first sentence “nonspecific” should be removed since chronic neck pain refers the duration and nonspecific refers the etiology.

Nonspecific was removed.

Methods: Even if the Institutional Ethics Committee has been named, is necessary to include the identification number or date of approval. In addition, this should be added in line 342-343

Identification number added.

CONSORT is a guideline, not statement. Reference is necessary

Corrected.

“In this study, the assessor and patients were blinded, but patients were aware of what treatment they were participating in”. In my opinion and based on this sentence, this study is not double-blinded since participants were aware of the treatment group.

Corrected.

Results: My request was not just to provide the number/percentage of males and females in each group. A new column with two groups (males and females) should be added reporting sociodemographic and clinical characteristics and analyzing if there are between-genders difference by using a Student-T test if variables are normal distributed or non-parametric test if not.

P- value was added for gender differences.